# Surgical Strategy for the Management of Cervical Deformity Is Based on Type of Cervical Deformity

**DOI:** 10.3390/jcm10214826

**Published:** 2021-10-21

**Authors:** Han Jo Kim, Sohrab Virk, Jonathan Elysee, Christopher Ames, Peter Passias, Christopher Shaffrey, Gregory Mundis, Themistocles Protopsaltis, Munish Gupta, Eric Klineberg, Robert Hart, Justin S. Smith, Shay Bess, Frank Schwab, Renaud Lafage, Virginie Lafage

**Affiliations:** 1Department of Orthopedics, Hospital for Special Surgery, New York, NY 10021, USA; elyseej@hss.edu (J.E.); fschwab@att.net (F.S.); renaud.lafage@gmail.com (R.L.); virginie.lafage@gmail.com (V.L.); 2Department of Orthopedics, Northwell Health, Great Neck, New York, NY 11021, USA; svirk@Northwell.edu; 3Department of Neurosurgery, University of San Francisco School of Medicine, San Francisco, CA 94143, USA; Christopher.Ames@ucsf.edu; 4Department of Orthopedics, NYU Langone Orthopedic Hospital, New York, NY 10016, USA; pgpassias@yahoo.com (P.P.); tprotopsaltis@gmail.com (T.P.); 5Department of Neurosurgery, Duke University Medical Center, Durham, NC 27708, USA; christopher.shaffrey@duke.edu; 6Division of Orthopaedic Surgery, Scripps Clinic Medical Group, La Jolla, CA 92037, USA; gmundis1@gmail.com; 7Department of Orthopaedic Surgery, Washington University, St. Louis, MO 63010, USA; munishgupta@wustl.edu; 8Department of Orthopedic Surgery, University of California Davis, Davis, CA 95616, USA; eoklineberg@ucdavis.edu; 9Department of Orthopaedic Surgery, Oregon Health & Science University, Portland, OR 97239, USA; robert.hart@swedish.org; 10Department of Neurosurgery, University of Virginia Medical Center, Charlottesville, VA 22904, USA; JSS7F@hscmail.mcc.virginia.edu; 11Denver International Spine Center, Rocky Mountain Hospital for Children at Presbyterian St. Luke’s, Denver, CO 80218, USA; shay_bess@hotmail.com

**Keywords:** cervical deformity, adult spinal deformity, surgical technique, surgical strategy, cervical osteotomy

## Abstract

Objectives: Cervical deformity morphotypes based on type and location of deformity have previously been described. This study aimed to examine the surgical strategies implemented to treat these deformity types and identify if differences in treatment strategies impact surgical outcomes. Our hypothesis was that surgical strategies will differ based on different morphologies of cervical deformity. Methods: Adult patients enrolled in a prospective cervical deformity database were classified into four deformity types (Flatneck (FN), Focal kyphosis (FK), Cervicothoracic kyphosis (CTK) and Coronal (C)), as previously described. We analyzed group differences in demographics, preoperative symptoms, health-related quality of life scores (HRQOLs), and surgical strategies were evaluated, and postop radiographic and HROQLs at 1+ year follow up were compared. Results: 90/109 eligible patients (mean age 63.3 ± 9.2, 64% female, CCI 1.01 ± 1.36) were evaluated. Group distributions included FN = 33%, FK = 29%, CTK = 29%, and C = 9%. Significant differences were noted in the surgical approaches for the four types of deformities, with FN and FK having a high number of anterior/posterior (APSF) approaches, while CTK and C had more posterior only (PSF) approaches. For FN and FK, PSF was utilized more in cases with prior anterior surgery (70% vs. 25%). For FN group, PSF resulted in inferior neck disability index compared to those receiving APSF suggesting APSF is superior for FN types. CTK types had more three-column osteotomies (3CO) (*p* < 0.01) and longer fusions with the LIV below T7 (*p* < 0.01). There were no differences in the UIV between all deformity types (*p* = 0.19). All four types of deformities had significant improvement in NRS neck pain post-op (*p* < 0.05) with their respective surgical strategies. Conclusions: The four types of cervical deformities had different surgical strategies to achieve improvements in HRQOLs. FN and FK types were more often treated with APSF surgery, while types CTK and C were more likely to undergo PSF. CTK deformities had the highest number of 3COs. This information may provide guidelines for the successful management of cervical deformities.

## 1. Introduction

The cervical spine is under substantial physiologic demands to allow for range of motion, maintain horizontal gaze, as well as support the weight of the cranium. For patients suffering from cervical deformity, all three of these functions are impeded. As a result, patients suffering from cervical deformity represent an extremely debilitated cohort of patients [1,2]. Surgeries to improve the alignment for this patient population, however, is not simple and can be associated with significant complications [3,4]. Although surgery may be associated with complications, patients still benefit significantly from appropriately performed procedures [5,6,7].

A layer of complexity is added for patients with cervical deformity given the wide range of radiographic presentations they have. The driver of their deformity can be from the cervical spine, cervicothoracic spine, thoracic spine or from spinopelvic alignment [8,9]. Recently, there has been an emphasis on using a data-driven approach to define subtypes of cervical deformity that present in similar patterns. Using this methodology, three sagittal morphotypes of cervical deformity have been identified: flatneck (FN), focal kyphosis (FK), and cervicothoracic (CTK) [10]. Coronal cervical deformity has also been identified as a unique clinical entity for patients with cervical deformity (C) [11]. FN patients have a large thoracic slope-cervical lordosis (TS-CL) while maintaining some ability to compensate for their deformity with extension. The FK subtype demonstrates a large focal kyphosis without necessarily having large global deformity. CT patients tend to have a large T1 slope with a large amount of cervical lordosis in an attempt to compensate for the deformity driven from more distal segments of the spine. Finally, the C entity represents patients with a coronal deformity without necessarily having a significant sagittal deformity.

The objective of our current study is to describe the surgical strategy for each subtype of cervical deformity. We hypothesized that there would be specific treatment patterns for each type of cervical deformity (FN, FK, CTK, and C). Furthermore, the second aim was to investigate if there were unique health related quality of life (HRQOL) patterns and radiographic parameters that were specific to each subtype of cervical deformity.

## 2. Method

### 2.1. Patient Population

We performed a retrospective review of a prospectively collected multi-center database. Patients were enrolled into the database across 13 sites around the United States between 2012–2015. The study was Institutional Review Board (IRB)-approved at each site, and patients signed a consent form prior to enrollment. Included patients in the database had to be over 18 years old and have met one of the following criteria: cervical scoliosis greater than 10 degrees, cervical kyphosis over 10 degrees, cervical sagittal certical axis (cSVA) over 4 cm, or a chin-brow vertical angle (CBVA) over 25 degrees.

### 2.2. Data Collection

We collected basic demographic information for each patient, including age, gender, body mass index (BMI), and Charlson comorbidities index (CCI). Health-related quality of life scores (HRQOLs) were collected for patients at the last follow-up visit, which was at least 1 year out from surgery. These included the numeric rating scale (NRS) back and neck, modified Japanese Orthopedic Association score (mJOA), EuroQual-5D (EQ-5D0), and neck disability index (NDI) for each patient.

Measurements were collected for both spinopelvic and cervical parameters. Specifically for spinopelvic parameters we measured pelvic incidence (PI), pelvic tilt (PT), lumbar lordosis (LL), PI-LL, T2-T12 sagittal cobb angle, T1 spinopelvic inclination (T1SPi), T1 pelvic angle (TPA), and sagittal vertical axis (SVA), cervical parameters were collected on flexion, extension, and neutral radiographs. These included C2-T3 segmental sagittal and coronal cobb angles, segmental Harrison angles, T1 slope (TS), C2–C7 sagittal cobb angle, TS-CL, cervical sagittal vertical axis (cSVA), and C2 slope. These measurements were made on both full-spine (36 inch minimum) radiographs and cervical radiographs. A schematic representing a portion of these measurements is shown in Figure 1.

Surgical information was collected for each patient. This included the approach for surgery (anterior, posterior or a combined approach), upper instrumented vertebra (UIV), lower instrumented vertebra (LIV), osteotomy, and location of the osteotomy. Three column osteotomies (3CO) were also recorded. HRQOLs and radiographic parameters were collected before and after surgical treatment. These HRQOLs included NRS-neck, NRS-arm, mJOA, EQ-5D, and NDI.

## 3. Statistical Analysis

We first listed descriptions of our entire cohort in terms of HRQOLs and radiographic parameters. The prevalence of each deformity subtype was also listed for each patient. These subtypes included Type 1: flatneck (FN), Type 2: focal kyphosis (FK), Type 3: cervicothoracic (CTK), and type 4: coronal (C), and have been previously described in the literature [7,10]. A schematic showing the four distinct morphotypes is shown in Figure 2.

We stratified our pre-operative data, surgical information, and post-operative outcomes for each subtype of cervical deformity (CD). We compared surgical strategies between the four subtypes. This included comparing approach of surgery, UIV selection, LIV selection and osteotomy type. When possible, a sub-category analysis was performed within each cervical deformity to compare surgical strategies, A paired t-test was used to compare pre- and post-surgical treatment continuous variables. A chi-square and exact Fisher test were used to compare surgical strategy between type of deformity. A *p* value of 0.05 was considered significant.

## 4. Results

A total of 90 patients were included within our analysis. No patients were excluded from this cohort. The mean age was 63.3 ± 9.2 years old. The mean BMI was 28.7 ± 7.2 kg/M^2^. There were no statistically significant differences in BMI between the types of cervical deformity (*p* > 0.05). There were more females than males (64% females). The mean CCI was 1.01 ± 1.36. Patients were subcategorized into cervical deformity types. There were 8 patients in the coronal (C) group (8.9%), 26 in the focal kyphosis (FK) group (28.9%), 30 in the flatneck (FN) group (33.3%), and 26 in the cervicothoracic (CT) group (28.9%).

### 4.1. Type 1: Flatneck Deformity

The mean age for the FN cohort was 65.1 ± 9.0 years old. The majority were female (60.0%). The mean BMI was 28.7 ± 8.2, and there was a significant number of patients in this cohort that were revision cases (53.3%, *N* = 16).

Pre-operative data for the FN patients is shown in Table 1. The HRQOLs demonstrated a severe disability without neurologic impairment. Sagittal alignment showed acceptable range with an overall maintained global alignment. Pre-operative cervical alignment showed a large cervical mismatch (56.5°) due to a large kyphotic curve (−16.5°), despite having some reserve of extension (14.9°).

The majority of FN patients were treated with a posterior only approach (56.7%), while a significant number were treated with a combined anterior approach (40%). One patient was treated with an anterior alone approach. There were five patients treated with a 3CO. The surgery location was in the upper thoracic spine (53.3%) between T1-T4 and 13.3% had a surgery with a LIV at T10 or lower.

Post-operative outcomes for FN patients are shown in Table 1**.** There was a significant decrease in neck pain post-surgery (*p* = 0.001). There was a significant increase in TPA (*p* = 0.006) and SVA (*p* = 0.027). No other significant changes were found in thoracolumbar parameters, although there was a trend toward higher PI-LL (*p* = 0.059). There was a significant reduction in cervical mismatch (Δ = −18.4 *p* = 0.001) due to the reduction in C2–C7 kyphosis (Δ22.7 *p* = 0.002) despite no significant difference in T1 slope (*p* = 0.237). At the time of our analysis, there were six patients (20.0%) that required a revision surgery. There were two patients that had residual cervical stenosis post-operatively. One patient had symptomatic pseudarthrosis. Another patient had multiple compression fractures in the thoracic spine requiring another spine surgery. One patient developed distal junctional kyphosis, and another patient developed proximal junctional failure.

A sub-analysis was performed to compare posterior only versus combined approaches for surgical correction. Only T1S was significantly different pre-op (44° ± 15° for posterior only vs. 29° ± 6° for combined approaches, *p* = 0.002), but other parameters were not significantly different (all *p* > 0.05). Patients that were revision cases were more likely to be treated with a posterior alone approach (70% vs. 25% *p* = 0.025). The mJOA scores for higher for those patients treated with a posterior alone approach (mJOA: 12.9 ± 1.8 vs. 15.2 ± 2.2 *p* = 0.007). Difference in mJOA remained significant post-op (13.3 ± 2.5 vs. 15.7 ± 1.9 *p* = 0.034) as well as higher disability post-op for posterior only (NDI: 52.2 ± 15.7 vs. 36.7 ± 18.8 *p* = 0.035). There was no significant difference in revision rate between the two surgical strategies.

### 4.2. Type 2: Focal Kyphosis

The mean age for the focal kyphosis (FK) cohort was 61.6 ± 7.0 years old. The majority of patients were female (77%). The mean BMI was 26.9 ± 6.0 kg/M^2^. There was a significant sub-group of patients that were revision cases (30.8%, *N* = 8).

Pre-operative data for the FK cohort is shown in Table 2. The pre-operative HRQOL scores did show myelopathic symptoms (mJOA) combined with severe disability (high NDI). Thoracolumbar alignment was not impaired for this cohort. Cervical alignment showed a larger focal kyphosis between two adjacent segments (−19.0° ± 10.0) with an overall maintained TS-CL mismatch due to a small T1 slope (19.4°).

The surgical approach utilized was fairly evenly split. The higher amount was a combined anterior and posterior approach (53.8%), and anterior only and posterior only both represented 23.1% of cases. A 3CO was used for three patients. For patients treated with an anterior only approach, the UIV was majority C3 (50%) and C4 (33.3%), and the LIV was majority C7 (83.3%). When a posterior or combined approach was used, the UIV was C2 in 70% of cases, and 65.0% had levels between C2 and T1-4.

Post-operative outcomes for the FK cohort are shown in Table 2**.** There was a significant improvement in neck pain (Δ = 1.4 *p* = 0.035), mJOA (Δ1.7 *p* = 0.034). There was also a trend toward improved NDI (*p* = 0.069) and EQ5D (*p* = 0.082). Post-op there was a significant increase in thoracic kyphosis (Δ = −6.7 *p* = 0.007) but no other significant change in global alignment. There was significant improvement in C2–C7 (Δ = 22.9 *p* < 0.001) and TS-CL (Δ = −16.8 *p* = 0.007) despite an increase in T1 slope (Δ6.1 *p* = 0.026). At the time of this analysis, 7.7% of patients had undergone revision surgery (*N* = 2). One patient had a post-operative infection requiring revision. Another patient required revision due to continued neck pain.

### 4.3. Type 3: Cervico-Thoracic Deformity

The mean age for the CT cohort was 64.8 ± 8.2 years old. The majority were females (62.0%), and the mean BMI was 30.4 ± 6.3 kg/M^2^. The majority of cases were revision cases (76.9%, *N* = 20).

Pre-operative data for the CTK cohort of patients is shown in Table 3. HRQOLs demonstrated severe disability without significant neurologic impairment. Sagittal alignment showed a large thoracic kyphosis (TK = 74°) combined with hyper extension of lordosis (PI-LL = 0) to maintain neutral global alignment (TPA = 15°, SVA = 6 mm). Pre-operative cervical alignment demonstrated a steep T1S and large cervical lordosis without a reserve of extension.

The majority of patients in the CTK cohort were treated with a posterior approach. A large portion (*N* = 11, 42.3%) were treated with a 3CO. The majority of UIV was located at C2 (34.6%), C3 (15.4%), or C4 (11.5%). The LIV was between T10–L2 for 42.3% of the patients and between T5–T9 for 34.6% of patients.

Post-operative outcomes for the CTK cohort are shown in Table 3**.** There were no significant changes in HRQOLs besides a trend for lower neck pain (*p* = 0.052). There was a significant reduction in thoracic kyphosis (*p* = 0.001) and a significant increase in PI-LL, TPA, and SVA (*p* < 0.01). There was a significant reduction in C2–T3 kyphosis (Δ = 29.1 *p* < 0.001), T1 Slope (Δ = −12.2 *p* < 0.001), and TS-CL (Δ = −22.9 *p* < 0.001) and a significant increase in C2–C7 (Δ = 11.8 *p* = 0.010). At the time of our analysis, there was a 19.2% (*N* = 5) rate of revision surgery. One patient had multiple compression fractures in the thoracic spine requiring a revision procedure. One patient required a revision for new onset weakness from cervical stenosis. One other patient developed distal junctional kyphosis requiring revision. Finally, one patient required revision due to pseudarthrosis.

A sub-analysis was performed on whether or not a 3CO was performed within the CTK cohort. There were no significant differences in pre-operative or post-operative alignment (all *p* > 0.05). There was a larger pre-operative NDI associated with patients that required a 3CO (43 ± 14 vs. 56 ± 13 *p* = 0.027). There was a trend towards a lower revision rate for the patients treated with a 3CO (*p* = 0.053).

### 4.4. Type 4: Coronal Deformity

The mean age for the C group was 57.5 ± 15 years old, and 42.9% were female. The mean BMI was 28.5 ± 9.4 kg/M^2^. There was one revision case within this sub-category of cervical deformity. Pre-operative data for this cohort is shown in Table 4. HRQOLs from this cohort demonstrate severe disability and pain without neurologic impairment. While sagittal alignment demonstrated acceptable values, there were significant issues with coronal alignment. There was a large upper thoracic cobb angle (45.8° ± 21.4) and a significant cervical curve (39.0° ± 16.0°).

The surgical treatment for C patients was mostly posterior only (*N* = 6, 62.0% of C patients). There were three patients treated with a combined anterior-posterior approach (*N* = 3, 37.5%). The UIV was mainly C2 (62.5%, *N* = 6). The LIV was primarily upper thoracic (T1–T4, 50%, *N* = 4) or mid-thoracic (T5–T9, 25%, *N* = 2). Post-operative outcomes are also shown in Table 4. There were significant reductions in neck pain (*p* = 0.004) and a trend for decreased back pain (*p* = 0.067). There were no significant changes in terms of mJOA, NDI, or EQ5D. The radiographic alignments showed that only TPA (*p* = 0.035) and SVA (*p* = 0.010) had a statistically significant change for spino-pelvic parameters. There were significant reductions in upper thoracic coronal cobb angle (Δ = −28.9 ± 14.9 *p* = 0.030) and cervical coronal cobb (Δ = 22.4° ± 7.3 *p* < 0.001). At the time of this data analysis, there were no revisions within our cohort of patients.

### 4.5. Comparison between Deformity Types

We performed a comparison across deformity types for approach, 3CO, UIV, and LIV treated. Type 2 (FK) was the only type treated with an anterior only approach, and there were also more combined approaches for FK patients (post hoc *p* = 0.007). A comparison of posterior only versus a combined approach showed that type 3 (CTK) were more commonly treated with a posterior alone approach (post hoc *p* = 0.017), with a post hoc analysis showing that a 3CO in the thoracic spine was more likely to be used for the treatment of type 3 CTK patients. There was a significant difference in the rate of 3CO across the subtypes of cervical deformity (*p* = 0.022). The UIV selected did not significantly vary across cervical subtypes. There was, however, a significant difference in the LIV selected across subtypes (*p* < 0.001). A post hoc analysis showed that type 3 CTK patients had less upper thoracic LIV (*p* = 0.006) and more thoracolumbar LIV (*p* < 0.001). For patients with type 2 FK, there were significantly more patients with a LIV in the cervical spine (*p* = 0002). A portion of our results are outlined in Table 5.

## 5. Discussion

We have found that different pre-operative alignment patterns lead to different surgical strategies for correction. Type 3 CTK patients were treated with a longer constructs (with an LIV into the mid-thoracic, lower thoracic or upper lumbar spine) with a higher rate of 3CO using a posterior only approach. Type 2 FK patients were treated with shorter constructs (LIV into C7, T1, T2) that often required both anterior/posterior approaches. Type 1 FN patients, however, had a more heterogeneous approach for treatment and a lower number of 3CO compared to type 3 and LIV that were more common in the upper thoracic spine (T2, T3, T4) than in the lower thoracic/upper lumbar spine (as seen in the type 3 patients). The variability seen in type 1s, however, is likely due to the need to both improve horizontal gaze, high rate of pre-operative revisions (+ 50% of cases) and correct any focal kyphosis present [12]. The C deformity was a rare presentation that prevented us from performing an in-depth analysis of surgical treatment/outcomes.

There was a significant rate of 3CO performed for our cohort of cervical deformity patients. Previous studies have shown the benefit of 3CO in cervical deformity to correct alignment [13]. Continuous improvements in techniques have also made the procedure safer [14,15]. Theologis et al. showed that 3CO can be used in the lower cervical and upper thoracic spine in order to gain significant correction [16]. It can also be performed in an efficient one-stage procedure [17]. Still, surgeons should use caution when employing this powerful tool. There is still over a 50% rate of at least one complication [13]. Given the rare nature of this indication, it can also be harder for surgeons to gain sufficient experience to safely perform this procedure [18]. Alternatively, employing multiple lower grade osteotomies may allow for significant correction with a lower overall risk of complications [19]. Our classification suggests 3CO may not be necessary except for in type 3 patients.

Our analysis provides valuable insight on a selection of approaches for patients with cervical deformity. There is a large variability among expert opinion on the best approach for most patients with cervical deformity [18]. Koller et al. has shown previously how different pre-operative sagittal balance types influence the surgical approach for treatment and how the degree of alignment changes for patients with rigid cervical deformity [20]. We have shown how a Type 2 FK deformity may be amenable to a combined approach whereas a Type 3 CTK deformity may be best treated with a posterior only approach and will likely involve the need for a 3CO. This is not surprising, since Type 3 patients typically have a very high T1S, for which a 3CO can be useful in correcting. Previous studies have also shown that local kyphosis may be more amenable to treatment with a combined approach and how a large deformity at the CTK junction may lend to treatment with a posterior approach [12,21]. Thus, we saw a higher rate of combined approaches in Type 2 deformities, but higher posterior only approaches in Type 3s. Combined approaches may allow for a higher rate of fusion, but it does come with additional risk, and surgeons should keep this in mind when treating patients with Type 2 FK [22]. There is a subset of patients, however, in which an anterior only approach may not be possible for cervical deformity, such as that seen in type 2 patients [23].

The selection of LIV varied across cervical deformity subtypes. Previous research has offered guidance when selecting LIV for ankylosing spondylosis or scheurman’s kyphosis, but there are limited data available for cervical deformity patients [24,25]. Previous literature has indicated that longer constructs with >9 levels of fusion are predictors of poor post-operative outcomes [26]. They have also been associated with increased operating room times, estimated blood loss, and length of stay [27]. Ultimately, however, the fusion length will also depend on the magnitude of the deformity, the location of the deformity, and presence/absence of concurrent degeneration at the adjacent segments in the planned end vertebrae. Larger studies are required to provide further insight on this complex clinical question.

There are several important limitations to our current study. This is a retrospective study and does not include an intent to treat analysis; nor did we take into account the methodology of pre-operative planning for the cases analyzed. In other words, we did not attempt to quantify the decision-making process for the surgical strategy attempted for each patient. This was difficult to assess due to the variability involved in surgical treatment strategies amongst different surgeons. Surgical strategy likely cannot be simplified to a few options (approach, osteotomy, fusion length, etc.), and larger studies are required to investigate such things as intra-op traction, instrumentation type, graft material, etc. We also have short radiographic follow up, and there is a potential for further deterioration in terms of long term follow up. This is particularly important when considering distal junctional kyphosis. However, we believe that this classification can provide a framework for the treatment of cervical deformity patient when it comes to level selection and surgical strategy. We also did not examine complications and how complications may vary based on the type of cervical deformity with which a patient would be dealing. Future research on, for example, the difference in complication rates between approaches for each type of cervical deformity may provide surgeons with useful information on how to treat patients with cervical deformity. Our study did not examine the minimally clinical important difference (MCID) for patients with a cervical deformity. It is difficult for us to use MCID scores for patients with a cervical deformity. As has been shown by Smith et al., patients with a cervical deformity are extremely disabled [1]. Using a MCID value that would be used for patients with, for example, cervical radiculopathy may not be valid. Further research on what an appropriate MCID value would be for cervical deformity should be found to help surgeons/researchers define an optimum/successful surgical strategy for these difficult patients. Finally, our follow up time for our study is only 1 year at minimum, and may be too short to adequately follow long term results of patients with cervical deformity.

In conclusion, we have shown the varied surgical treatment strategies and outcomes associated with each cervical deformity subtype. As with thoracolumbar deformity, cervical deformity is a broad term, and each surgical strategy will likely require a patient-specific analysis. Our current results can serve as a basis to think through this surgical decision-making process.

## Figures and Tables

**Figure 1 jcm-10-04826-f001:**
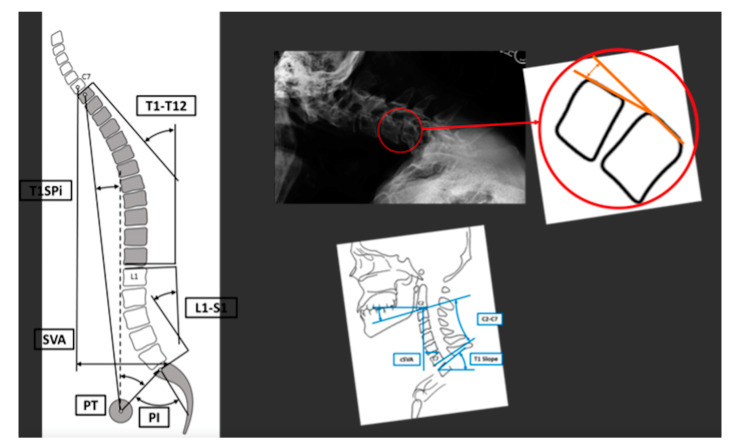
These schematics show a portion of the various radiographic measurements that were recorded for each patient. (T1SPi = T1 spinopelvic inclination, SVA = sagittal vertical axis, cSVA = cervical sagittal vertical axis).

**Figure 2 jcm-10-04826-f002:**
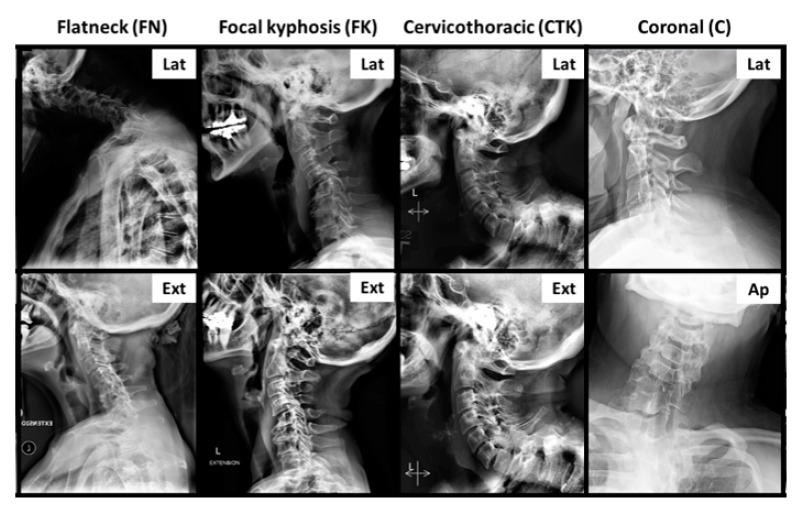
The four morphotypes of cervical deformity are shown here. “Lat” refers to a neutral lateral radiograph. “Ext” represents an extended neck lateral radiograph.

**Table 1 jcm-10-04826-t001:** Pre-operative and post-operative patient reported outcomes and radiographic sagittal alignment for patients with a Type 1—Flatneck (FN).

HRQOL		**NSR Back**	**NSR Neck**	**mJOA**	**EQ5D**	**NDI**	
	Pre	4.7 ± 2.9	6.6 ± 2.5	13.8 ± 2.2	0.7 ± 0.1	45.9 ± 18.5	
	Post	5.5 ± 2.4	4.6 ± 2.7	14 ± 2.6	0.7 ± 0.1	46 ± 18.3	
	*p*-value	0.940	0.001	0.780	0.605	0.952	
Neutral x-ray		**PI**	**PT**	**PI-LL**	**T2-T12**	**TPA**	**SVA**
	Pre	55.3 ± 9.8	20.5 ± 9.8	−0.9 ± 13.9	−56.5 ± 18.4	13.8 ± 9.6	1 ± 70
	Post	55 ± 10.8	22.9 ± 10.7	4 ± 14	−63.6 ± 17.3	19.5 ± 12.6	38 ± 83.4
	*p*-value	0.509	0.314	0.059	0.185	0.006	0.027
		**C2-T3**	**T1 Slope**	**C2-C7**	**TS-CL**	**cSVA**	**C2 Slope**
	Pre	−29.5 ± 22.2	38.2 ± 14.3	−16.5 ± 22.9	56.5 ± 18.8	68.3 ± 15.2	53.6 ± 17.9
	Post	−1.4 ± 14.2	44.9 ± 19.8	10.8 ± 15.8	36.6 ± 19.3	53.5 ± 15.1	35.6 ± 18.7
	*p*-value	0.000	0.237	0.000	0.000	0.001	0.000
Dynamic X-ray		**C2-C7 Ext.**	**TS-CL Ext.**	**C2-C7 Flex.**	**TS-CL Flex.**	**C2-C7 Res.**	**TS-CL Res.**
	Pre	−1.6 ± 25	34.9 ± 22.9	−27.3 ± 24	76.2 ± 20.8	14.9 ± 10.5	−21.7 ± 12.3

**Table 2 jcm-10-04826-t002:** Pre-operative and post-operative patient reported outcomes and radiographic sagittal alignment for patients with a Type 2—Focal Kyphosis (FK).

HRQOL		**NSR Back**	**NSR Neck**	**mJOA**	**EQ5D**	**NDI**	
	Pre	5 ± 2.8	6 ± 2.5	12.2 ± 3.3	0.7 ± 0.1	46.4 ± 15.6	
	Post	4.1 ± 3.2	4.6 ± 2.9	13.8 ± 2.8	0.8 ± 0.1	41.2 ± 17.6	
	*p*-value	0.120	0.035	0.034	0.082	0.069	
Neutral x-ray		**PI**	**PT**	**PI-LL**	**T2-T12**	**TPA**	**SVA**
	Pre	52 ± 13.2	19.3 ± 11.6	1.4 ± 19.6	−39.2 ± 16.9	12.8 ± 12	−9 ± 63
	Post	50.5 ± 12.7	18.8 ± 10.2	−0.6 ± 17.7	−48 ± 18.5	14.2 ± 10.8	12.8 ± 61.5
	*p*-value	0.844	0.528	0.832	0.009	0.068	0.236
		**C2-T3**	**T1 Slope**	**C2-C7**	**TS-CL**	**cSVA**	**C2 Slope**
	Pre	−19.7 ± 25.1	19.4 ± 16.4	−12.2 ± 23.2	31.8 ± 15.2	35.3 ± 25.2	36.1 ± 26.4
	Post	1.4 ± 10.7	28.9 ± 16.4	6.5 ± 11.7	22.8 ± 8.5	30.7 ± 19.4	23.1 ± 12.1
	*p*-value	0.003	0.026	0.000	0.007	0.107	0.019
Dynamic X-ray		**C2-C7 Ext.**	**TS-CL Ext.**	**C2-C7 Flex.**	**TS-CL Flex.**	**C2-C7 Res.**	**TS-CL Res.**
	Pre	−0.2 ± 19.6	17.2 ± 9.9	−28.9 ± 16.1	58.4 ± 14.6	7.9 ± 8.4	−10.3 ± 8.8

**Table 3 jcm-10-04826-t003:** Pre-operative and post-operative patient reported outcomes and radiographic sagittal alignment for patients with a Type 3—Cervicothoracic (CT).

HRQOL		**NSR Back**	**NSR Neck**	**mJOA**	**EQ5D**	**NDI**	
	Pre	5 ± 2.8	7 ± 2.3	13.9 ± 3	0.7 ± 0.1	48.5 ± 14.9	
	Post	5.3 ± 3.4	6.1 ± 2.8	14.2 ± 2.5	0.7 ± 0.1	46.8 ± 19.9	
	*p*-value	0.951	0.052	0.770	0.460	0.498	
Neutral x-ray		**PI**	**PT**	**PI-LL**	**T2-T12**	**TPA**	**SVA**
	Pre	56.3 ± 11.8	22.8 ± 11.9	−0.1 ± 20.5	−74 ± 20	15.3 ± 13	6 ± 70
	Post	57 ± 12.4	23.6 ± 12	5.4 ± 19.6	−62.3 ± 16.7	18.9 ± 13.1	34.3 ± 67.9
	*p*-value	0.954	0.903	0.011	0.001	0.010	0.001
		**C2-T3**	**T1 Slope**	**C2-C7**	**TS-CL**	**cSVA**	**C2 Slope**
	Pre	−22.3 ± 27.4	56.4 ± 13.9	9.1 ± 22.4	49.6 ± 19.1	66.1 ± 12.7	50.7 ± 20.1
	Post	7 ± 15.5	46 ± 16.5	20.2 ± 18.3	26.3 ± 13.7	45.4 ± 12.7	23.3 ± 12.4
	*p*-value	0.000	0.001	0.010	0.000	0.000	0.000
Dynamic X-ray		**C2-C7 Ext.**	**TS-CL Ext.**	**C2-C7 Flex.**	**TS-CL Flex.**	**C2-C7 Res.**	**TS-CL Res.**
	Pre	9.9 ± 23.6	45.4 ± 19.5	−4.5 ± 20.5	67.9 ± 15.6	0 ± 7.4	−2.7 ± 8

**Table 4 jcm-10-04826-t004:** Pre-operative and post-operative patient reported outcomes and radiographic sagittal alignment for patients with a Type 4—Coronal (C).

HRQOL		**NSR Back**	**NSR Neck**	**mJOA**	**EQ5D**	**NDI**	
	Pre	6 ± 3.2	6 ± 2.4	12.6 ± 3.4	0.7 ± 0.1	52.4 ± 22.1	
	Post	3.6 ± 3	3.1 ± 2.4	13.6 ± 5	0.7 ± 0.1	37.7 ± 23	
	*p*-value	0.067	0.004	0.642	0.677	0.222	
Neutral x-ray		**PI**	**PT**	**PI-LL**	**T2-T12**	**TPA**	**SVA**
	Pre	55.1 ± 11.3	19.3 ± 15.3	3.8 ± 26.2	−40.6 ± 17.2	12.4 ± 18.7	−14 ± 72
	Post	55.4 ± 12.2	25.7 ± 18.7	12.3 ± 30.7	−43.8 ± 21.5	21.7 ± 22.5	19.6 ± 77.8
	*p*-value	0.766	0.152	0.139	0.408	0.035	0.010
		**C2-T3**	**T1 Slope**	**C2-C7**	**TS-CL**	**cSVA**	**C2 Slope**
	Pre	−12.4 ± 17.2	26.7 ± 9.6	−2.4 ± 10.6	32.5 ± 23.1	35.7 ± 21.1	26.6 ± 22.2
	Post	2.6 ± 17.2	34.2 ± 18.1	9.7 ± 16.8	27.4 ± 7.5	35.6 ± 15.3	20.6 ± 10.1
	*p*-value	0.220	0.242	0.227	0.602	0.553	0.361
Dynamic X-ray		**C2-C7 Ext.**	**TS-CL Ext.**	**C2-C7 Flex.**	**TS-CL Flex.**	**C2-C7 Res.**	**TS-CL Res.**
	Pre	7.8 ± 17.5	21.8 ± 21.4	−16.2 ± 13.7	54 ± 17.9	10.8 ± 19.7	−15.4 ± 13.7

**Table 5 jcm-10-04826-t005:** The breakdown of our analysis of operative treatment across deformity types is shown. There is a significant difference in the rate of 3CO and LIV selection across deformity groups. (FN = flatneck, FK = docal deformity, CTK = cervicothoracic, C = coronal, 3CO = three column osteotomy, LIV = lowest instrumented vertebra).

	C	FK	CTK	FN	*p* Value
Rate of 3CO	2/8 (25%)	3/26 (11.5%)	11/26 (42.3%)	5/30 (16.6%)	0.02
Rate of use of UT for LIV	4/8 (50%)	13/26 (50%)	2/26 (7.7%)	16/30 (53.3%)	<0.001

## Data Availability

Data available on request due to restrictions e.g., privacy or ethical. The data presented in this study are available on request from the corresponding author.

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
