# Peer review of "Surgical Strategy for the Management of Cervical Deformity Is Based on Type of Cervical Deformity"

_jcm, 2021, doi:10.3390/jcm10214826_

Round 1
Reviewer 1 Report
Dear Authors,
This is a well-designed and very useful study. many aspects of the topic had been exposed in an appropriate way. Language, expect few spelling errors, is good for a scientific paper.
Study design: multi-center, retrospective study.
Aim: The objective of the study was to describe the surgical strategy of cervical deformities divided in four types. Second (?) objective was to detect the QoL and radiographic parameters specific for each type of studied deformity.
Population: patients 18 y. or older, with cervical scoliosis greater than 10°. Cervical kyphosis greater than 10°, cervical sagittal 65 cervical axis over 4 centimeters or a chin-brow vertical angle over 25°.
Intervention: anterior, posterior or a combined approach
Collection: demographic, clinical and radiographic data
Outcome: cervical and spinal radiographic outcomes, QoL, demographic
Suggestions
The mean BMI is slightly higher in the groups. Is there a difference in QoL between normal (18.5-24.9), overweight (25-29.9) and above?
Abstract
Line 10: as previously described. Can you please explain to what do you refer?
Line 20: please write the full name NDI before the acronym.
Introduction
Line 53: please add of after “The objective”.
Line 55: please specify if radiographic parameters and HRQOL is second aim of the study.
Methods
Methods are clear and statistical analysis is inherent to the data and question and is well explained. I did not see the time the QoL was made. Can you please write it in the methods? Thank you.
Figure 2: please specify in the caption what lat and ext mean. Furthermore, are you able to add radiographs of flection as well? Thank you.
Results
Please specify why you excluded 19 patients.
Line 106-110: please specify why you excluded 19 patients.
Line 232: in the caption where written FK, please correct the spelling.
Discussion
It is well made. It explains the different strategies and the different pro and cons of each one. Nevertheless, I will stress more out, in another paragraphs, the QoL scores.
Conclusion
The conclusion is concise and supported by the results.
Reviewer 2 Report
Overall, I believe there are many informations missing in order to provide the reader with an actual surgical strategy:
- what was the decision process behind the choice of the kind of approach/extent of surgery...
-were there intraoperative complications? Did the rate of complications differ in the various approaches?
-The comparison between different approaches is very limited
-most PROMs barely/do not reach the minimal clinically important difference. Was the intention that of preventing worsening of the condition? Is surgery still seen as succesfull despite similar postoperative clinical condition?
-the follow-up is too short to determine whether a given approach is safe and leads to satisfactory results.
I am aware that the authors mention many of these issues in the limitations, however I think that leaving these analysis out does not offer guidance for the reader in the planning of the surgical approach.
Round 2
Reviewer 2 Report
I commend the authors on their work. However, I do not think that the limited follow-up is adequate to say whether the employed strategy was succesful or not. Also, the question on how success is measured still remains open: is the improvement in HRQOL sufficient? Was surgery seen as damage-control? Also, I believe that only enouncing the type of surgery performed, with multiple options presented in some cases, and without describing the decision-making process, would not be enough to guide other professionals in their choice of Approach.
Author Response
Dear Reviewer of Journal of Clinical Medicine,
Thank you for taking the time to review our article titled “Surgical Strategy for the Management of Cervical Deformity is based on type of Cervical deformity”. Please find below our answers to the points you have listed.
Comment:
However, I do not think that the limited follow-up is adequate to say whether the employed strategy was succesful or not. Also, the question on how success is measured still remains open: is the improvement in HRQOL sufficient? Was surgery seen as damage-control?
Reply:
Thank you for this comment. We agree with your sentiment regarding the follow-up time and we have listed the fact that we are not defining success in cervical deformity surgery as a limitation as stated below.
“Our study did not examine the minimally clinical important difference (MCID) for patients with cervical deformity. It is difficult for us to use MCID scores for patients with cervical deformity. As has been shown by Smith et al., patients with cervical deformity are extremely disabled1. Using a MCID value that would be used for patients with, for example, cervical radiculopathy may not be valid. Further research on what an appropriate MCID value would be for cervical deformity should be found to help surgeons/researchers define an optimum/successful surgical strategy for these difficult patients. Finally, our follow up time for our study is only 1 year at minimum and may be too short to adequately follow long term results of patients with cervical deformity.”
Once again, thank you for taking the time to review our manuscript. Please let us know if further edits are required.
Sincerely,
Sohrab Virk and Co-Authors
